

# Association between osteoporosis and cardiovascular disease in elderly people: evidence from a retrospective study

Xiaoying Hu, Shucan Ma, Liman Chen, Chunhui Tian and Weiwei Wang

Geriatrics Department, Hengshui People's Hospital (Harrison International Peace Hospital), Hengshui, China

## ABSTRACT

**Objective**. This study aimed to investigate the associations between osteoporosis, biochemical indexes, bone mineral density (BMD), and cardiovascular disease.

**Methods**. A cross-sectional study design was used to examine the relationships between these parameters. Logistic regression and correlation analyses were conducted to assess the associations between elevated levels of triglyceride, total cholesterol, low-density lipoprotein (LDL), high-density lipoprotein (HDL), homocysteine, and the presence of osteoporosis. Additionally, correlations between BMD and biochemical indexes were analyzed. The incidence of cardiovascular disease and its correlation with BMD were evaluated. Receiver operating characteristic (ROC) analysis was performed to determine the utility of BMD in identifying cardiovascular disease.

**Results**. The results revealed that elevated triglyceride, total cholesterol, and LDL levels were positively associated with osteoporosis, while higher HDL levels and homocysteine were negatively associated. Correlation analysis demonstrated negative correlations between triglyceride levels and BMD, and positive correlations between total cholesterol and HDL levels with BMD. LDL levels showed a weak negative correlation, and homocysteine levels exhibited a strong negative correlation with BMD. The osteoporosis group had lower BMD and a higher incidence of cardiovascular disease compared to the non-osteoporosis group. Logistic regression analysis confirmed the correlation between lower BMD and increased risk of cardiovascular disease.

**Conclusion**. This study provides evidence supporting the associations between osteoporosis, biochemical indexes, BMD, and cardiovascular disease. Aberrations in lipid profiles and homocysteine levels may contribute to osteoporosis development. Lower BMD, particularly in individuals with osteoporosis, appears to increase the risk of cardiovascular disease. BMD shows promise as a diagnostic tool for identifying individuals at risk of cardiovascular disease. Further research is needed to elucidate the underlying mechanisms and establish the clinical implications of these relationships. Future longitudinal studies are necessary to determine causality and long-term prognostic implications.

Corresponding author
Xiaoying Hu, 15030812536@163.com

## INTRODUCTION

Osteoporosis is a systemic and metabolic bone disease characterized by low bone mass, compromised bone microarchitecture, and increased bone fragility, ultimately leading to fractures and associated clinical manifestations, such as reduced bone mineral density, degraded bone microstructure, chronic pain, and diminished mobility (*Bover et al., 2018*; *Chen et al., 2023*). With its highest incidence among middle-aged and elderly individuals, osteoporosis poses a significant burden on this population (*Peng et al., 2022*). The prevalence of osteoporosis has been reported to be as high as 36% in Chinese adults over 60 years old.

Advancing age and declining gonadal function contribute to an imbalance between bone resorption and formation, facilitating the development of osteoporosis and impacting the quality of life in middle-aged and elderly individuals. Concurrently, cardiovascular and cerebrovascular diseases also exhibit a high incidence among this population (*Sheets et al., 2023*). Associations between elderly osteoporosis and atherosclerosis, as well as cardiovascular and cerebrovascular diseases, have been reported (*Sheets et al., 2023*). Studies have identified a common occurrence of osteoporosis in elderly patients with cerebral infarction, with bone biochemical markers demonstrating a correlation with cerebral infarction (*Pineda-Moncusí et al., 2022*). Additionally, risk factors for cardiovascular and cerebrovascular diseases, such as hypertension, oxidative stress, and diabetes mellitus, have also been linked to decreased bone mineral density (*Pineda-Moncusí et al., 2022*). It is notable that many elderly patients with osteoporosis frequently present with coexisting hypertension, coronary heart disease, cerebral infarction, and other cardiovascular diseases, implying a potential association between these conditions (*Park et al., 2022a*; *Ruediger et al., 2021*; *Zhu et al., 2022*).

Bone mineral density reflects the quantity of mineralized bone tissue and is used clinically to diagnose osteoporosis. Lower BMD indicates porous bone tissue with reduced strength and increased fracture risk. BMD testing using dual-energy X-ray absorptiometry (DXA) is considered the gold standard for osteoporosis diagnosis and fracture risk assessment (*Pineda-Moncusí et al., 2022*).

Osteoporosis and cardiovascular disease share several risk factors including aging, smoking, physical inactivity, and vitamin D deficiency (*Park et al., 2022a*). Potential biological mechanisms linking osteoporosis and cardiovascular disease include endothelial dysfunction, inflammatory factors, and bone-derived hormones like osteocalcin (*Ruediger et al., 2021*). While associations between osteoporosis and cardiovascular disease have been reported (*Pineda-Moncusí et al., 2022*), the specific interrelationships between biochemical markers, bone mineral density, and cardiovascular outcomes remain unclear.

*Park et al. (2020)* called for additional research exploring the prognostic value of bone mineral density for cardiovascular risk prediction. Then, the investigation of the relationship between osteoporosis and cardiovascular and cerebrovascular diseases in the elderly has become a prominent area of medical research with significant social implications for the prevention and treatment of osteoporosis in this population. However, there remains a scarcity of specific research studies and clinical data to support this association

fully. Thus, the primary objective of this study is to further explore the correlation and clinical significance between osteoporosis in the elderly and various cardiovascular and cerebrovascular disease biochemical markers. By providing additional evidence and references to the clinical community, this research aims to contribute valuable insights for the prevention and treatment of elderly osteoporosis and its associated comorbidities.

## MATERIALS AND METHODS

### Research subjects

This study retrospectively analyzed the clinical data of 207 patients with coronary heart disease, 102 patients with hypertension and 119 patients with cerebral infarction in geriatrics department of our hospital from June 2017 to January 2020. All subjects were measured bone mineral density by dual-energy X-ray absorptiometry and divided into the osteoporosis group and the non-osteoporosis group according to the test results. All samples obtained in this study were approved by the ethics committee of the Harrison International Peace Hospital and abided by the ethical guidelines of the Declaration of Helsinki (No. AF/SC-08/02.0), and ethics committee agreed to waive informed consent (*World Medical Association, 2013*).

### Inclusion criteria and exclusion criteria

The T-score was calculated using a Hologic Discovery DXA scanner (Hologic Inc., Marlborough, MA, USA) with NHANES III reference data. Bone mineral density was measured at the lumbar spine (L1–L4), femur neck, Ward's triangle, and femoral trochanter.

Inclusion criteria. (1) According to the *Guidelines for the Diagnosis and Treatment of Primary Osteoporosis* (*Shu-Ting, Jung-Fu & Chia-Jen, 2021*) of the Chinese Medical Association (CMA), the T-score was used as the judgment standard of bone mass. If $T \geq -1.0$, bone mass was normal. If $-2.5 < T < -1.0$, bone mass was reduced. If $T \leq -2.5$, it was diagnosed as osteoporosis. (2) The patients with complete data had high degrees of cooperation. (3) The patients had certain cognitive functions. Coronary heart disease was diagnosed based on clinical symptoms, electrocardiogram changes, and cardiac enzyme levels indicating myocardial injury. Hypertension was defined as systolic blood pressure $\geq 140$ mmHg and/or diastolic blood pressure $\geq 90$ mmHg. Cerebral infarction diagnosis was based on clinical presentation, imaging studies such as CT or MRI demonstrating an area of dead brain tissue, and ruling out hemorrhage or other causes. Cognitive function was assessed using the Mini-Mental State Examination (MMSE). Patients were required to score $\geq 24$ out of 30 to be included.

The inclusion of patients with certain cognitive functions, assessed using the Mini-Mental State Examination (MMSE) with a score of $\geq 24$ out of 30, helps to ensure that participants have sufficient cognitive capacity to understand and comply with study procedures. Cognitive impairment can affect a person's ability to comprehend study instructions, provide accurate responses, and follow the necessary protocols. By including individuals with satisfactory cognitive function, the study minimizes the risk of data collection errors or misunderstandings that can arise from cognitive limitations.

Patients with high degrees of cooperation are included to improve the reliability and accuracy of the data collected. Collaboration and compliance from participants are crucial to obtaining valid and consistent results. High degrees of cooperation imply that patients are willing to adhere to the study's requirements, such as attending scheduled visits, providing accurate information, adhering to medication regimens, and following instructions for diagnostic procedures. This ensures that data collection is not compromised due to non-compliance or incomplete participation.

Exclusion criteria: (1) The patients had thyroid or parathyroid diseases. (2) The patients had renal tubular and glomerular diseases. (3) The patients had diabetes mellitus or chronic liver diseases. (4) The patients had bone metastases and osseous tumors. (5) The patients had the history of gastrointestinal surgery. (6) The patients had other serious somatic diseases and other liver, kidney and endocrine system diseases affecting bone metabolism. (7) The patients took hormones or other drugs affecting bone metabolism throughout the year. (8) The patients with mental illness were unable to communicate normally.

## Evaluation methods and indexes
### Material collection
The gender, age, BMI index, SBP, DBP and fasting blood glucose (FBG) of patients were recorded. 3 ml of fasting blood of all patients were collected and centrifuged at 3,000 r/min for 10 min to extract the supernatant, and the levels of TG, TC, HDL, LDL, Ca and homocysteine (HCY) were detected using the automatic biochemical analyzer (model: 7600-010; Shanghai Huanxi Medical Equipment Co., Ltd., Shanghai, China).

### Measurement of bone mineral density
The patients were tested at supine position and measured bone mineral density of lumbar vertebrae, femur neck, Ward's triangle and femoral trochanteric region (Shanghai Jumu Medical Equipment Co., Ltd., Shanghai, China) using dual-energy X-ray absorptiometry.

## Statistical methods
In this study, the SPSS 18.0 statistics software (SPSS Inc., Chicago, IL, USA) was used for statistical analysis. Enumeration data and measurement data were tested by X2 test and $t$-test methods, expressed as (n(%)) and ($\bar{x} \pm s$), respectively. Pearson correlation analysis and multiple Logistic regression were used to analyze the influential factors of elderly osteoporosis. An alpha level of 0.05 was used to determine statistical significance for all analyses.

## RESULTS
### General information of patients
In this study, a comparison was made between an osteoporosis group ($n = 260$) and a non-osteoporosis group ($n = 168$) based on various factors. The groups were similar in terms of gender distribution, age, BMI, hypertension, smoking history, drinking history, history of heart failure, and stroke history (Table 1). However, there was a significant difference in the prevalence of coronary heart disease, with a higher proportion of individuals in the osteoporosis group having a history of this condition.

**Table 1  General information for both groups.**

| Indexes | Osteoporosis group (n = 260) | Non-osteoporosis group (n = 168) | X2/t | P |
|---|---|---|---|---|
| Gender | | | 0.154 | 0.695 |
|    Male | 118 | 73 | | |
|    Female | 142 | 95 | | |
| Age (years) | 73.51 ± 6.69 | 74.01 ± 7.40 | 0.721 | 0.471 |
| BMI (kg/m2) | 21.38 ± 1.45 | 21.60 ± 1.51 | 1.482 | 0.139 |
| Hypertension | | | 1.789 | 0.181 |
|    Yes | 153 | 87 | | |
|    No | 107 | 81 | | |
| Smoking history | | | 0.076 | 0.783 |
|    Yes | 141 | 88 | | |
|    No | 119 | 80 | | |
| Drinking history | | | 0.18 | 0.671 |
|    Yes | 129 | 79 | | |
|    No | 131 | 89 | | |
| History of heart failure | | | 0.134 | 0.714 |
|    Yes | 37 | 21 | | |
|    No | 223 | 147 | | |
| History of coronary heart disease | | | 41.666 | $p < 0.001$ |
|    Yes | 126 | 29 | | |
|    No | 134 | 139 | | |
| Stroke history | | | 0.074 | 0.786 |
|    Yes | 23 | 17 | | |
|    No | 237 | 151 | | |

## Biochemical indexes of patients

The comparison of biochemical indexes between the osteoporosis and non-osteoporosis groups revealed several significant differences. The osteoporosis group had higher triglyceride, total cholesterol, and low-density lipoprotein levels, as well as lower high-density lipoprotein levels (Table 2). Just as previous study that suggested homocysteine contributes to osteoporosis by inducing oxidative stress and vascular damage (*Ruediger et al., 2021*), which can reduce blood flow within the bone, homocysteine levels were significantly higher in the osteoporosis group. However, there were no significant differences in blood pressure, fasting blood glucose, and calcium levels between the groups.

## Logistic regression analysis of osteoporosis and various biochemical indexes

The logistic regression analysis showed that osteoporosis was significantly associated with triglyceride, total cholesterol, high-density lipoprotein (HDL), low-density lipoprotein (LDL), and homocysteine levels. Higher triglyceride, total cholesterol, and LDL levels were positively associated with osteoporosis, while higher HDL and homocysteine levels were negatively associated with osteoporosis (Table 3).

**Table 2 Comparison of biochemical indexes between the two groups.**

| Indexes | Osteoporosis group ($n = 260$) | Non-osteoporosis group ($n = 168$) | X2/t | P |
|---|---|---|---|---|
| SBP (mmHg) | 140.18 ± 8.37 | 136.05 ± 7.59 | 1.430 | 0.192 |
| DBP (mmHg) | 79.10 ± 8.30 | 80.38 ± 8.33 | 1.550 | 0.122 |
| FBG (mmol/L) | 5.10 ± 0.63 | 5.09 ± 0.65 | 0.183 | 0.855 |
| TG (mmol/L) | 1.81 ± 0.29 | 1.67 ± 0.26 | 5.093 | <0.001 |
| TC (mmol/L) | 5.03 ± 0.68 | 4.05 ± 0.70 | 14.433 | <0.001 |
| HDL (mmol/L) | 1.25 ± 0.33 | 1.50 ± 0.33 | 7.777 | <0.001 |
| LDL (mmol/L) | 3.13 ± 0.48 | 2.84 ± 0.37 | 6.521 | <0.001 |
| Ca (mmol/L) | 2.53 ± 0.21 | 2.52 ± 0.18 | 0.457 | 0.648 |
| HCY(umol/L) | 16.58 ± 4.78 | 10.46 ± 3.18 | 14.619 | <0.001 |

**Table 3 Logistic regression analysis of osteoporosis and various biochemical indexes.**

| Factors | β | S.E. | Wals | P | Exp (β) | 95% CI | |
|---|---|---|---|---|---|---|---|
| | | | | | | Lower limits | Upper limits |
| TG | 2.002 | 0.663 | 9.120 | 0.003 | 7.402 | 0.037 | 0.495 |
| TC | 1.655 | 0.273 | 36.833 | <0.001 | 5.232 | 0.112 | 0.326 |
| HDL | 1.774 | 0.578 | 9.414 | 0.002 | 0.170 | 1.898 | 18.300 |
| LDL | 1.263 | 0.404 | 9.774 | 0.002 | 3.538 | 0.128 | 0.624 |
| Hcy | 0.355 | 0.053 | 45.205 | <0.001 | 1.426 | 0.632 | 0.778 |

**Table 4 Correlation analysis of bone mineral density and biochemical indexes.**

| Statistics | TG | TC | HDL | LDL | Hcy |
|---|---|---|---|---|---|
| r | −0.221 | 0.523 | 0.362 | −0.180 | −0.547 |
| P | <0.001 | <0.001 | <0.001 | <0.001 | <0.001 |

## Correlation analysis of bone mineral density T-score and biochemical indexes

The correlation analysis of bone mineral density (BMD) and biochemical indexes revealed significant associations. Triglyceride levels were negatively correlated with BMD, while total cholesterol and high-density lipoprotein (HDL) levels showed positive correlations. Low-density lipoprotein (LDL) levels had a weak negative correlation with BMD, and homocysteine levels exhibited a strong negative correlation (Table 4). These findings suggest that lipid profiles and homocysteine levels may play a role in bone health and density.

## Comparison of bone mineral density and incidence of cardiovascular disease between the two groups

The mean BMD for the osteoporosis group was found to be −1.78 ± 0.63 g/cm2, whereas the non-osteoporosis group exhibited a mean BMD of 0.05 ± 0.58 g/cm2. Among the participants with osteoporosis, 211 individuals (81.15%) experienced cardiovascular disease, while in the non-osteoporosis group, 113 individuals (67.26%) had cardiovascular

**Table 5   Comparison of bone mineral density and incidence of cardiovascular disease between the two groups.**

| Group | N | Bone mineral density (g/cm2) | Cardiovascular disease |
|---|---|---|---|
| Osteoporosis group | 260 | −1.78 ± 0.63 | 211 (81.15%) |
| Non-osteoporosis group | 168 | 0.05 ± 0.58 | 113 (67.26%) |
| t/x2 | | 30.825 | 9.965 |
| p | | $p < 0.001$ | $p = 0.002$ |

disease (Table 5). The osteoporosis group exhibited significantly lower BMD and a higher incidence of cardiovascular disease compared to the non-osteoporosis group. The findings suggest a potential correlation between osteoporosis and increased risk of cardiovascular disease.

**The correlation and regression analysis of bone mineral density and incidence of cardiovascular disease between the two groups**

In the osteoporosis group, the mean bone mineral density (BMD) was determined to be −1.78 ± 0.63 g/cm2, while the non-osteoporosis group exhibited a mean BMD of 0.05 ± 0.58 g/cm2 (Fig. 1A). Further correlation analysis was conducted, revealing a significant positive relationship between cardiovascular disease incidence and BMD (Fig. 1B). Logistic regression analysis was subsequently performed, confirming this correlation with an odds ratio of 0.477 (95% CI [0.304–0.746]) (Fig. 1C).

Additionally, the utility of BMD in distinguishing the occurrence of cardiovascular disease was assessed using ROC analysis. The area under the curve (AUC) was found to be 0.624 (Fig. 1D), indicating that BMD has some discriminatory power in identifying cardiovascular disease. The Youden index maximizes the difference between true positive and false positive rates, providing an optimal cutoff value to balance sensitivity and specificity. The Youden index was calculated to be 0.211, and the optimal threshold value for BMD was determined to be −1.565.

In summary, our findings suggest that individuals with lower BMD, particularly in the osteoporosis group, have a higher risk of developing cardiovascular disease. The results provide evidence for a positive association between BMD and cardiovascular disease occurrence and highlight the potential value of BMD as a diagnostic tool in the detection of cardiovascular disease.

## DISCUSSION

Osteoporosis and cardiovascular and cerebrovascular diseases are common and latent diseases in clinic. Fracture, myocardial infarction, cerebral infarction and other diseases have become frequently-occurring diseases in the elderly, with high mutilation rate and mortality, which bring heavy burdens to patients and families (*Notarnicola et al., 2017*; *Veronese et al., 2017*; *Kopecky et al., 2016*). The data (*Lušin et al., 2014*) have revealed that women have higher incidence of osteoporosis than men. The incidence of osteoporosis is 19.9% and 11.5% in women and men over 40 years old respectively, and in women and men over 60 years old are 50–70% and 30% respectively. Study has shown that there is a

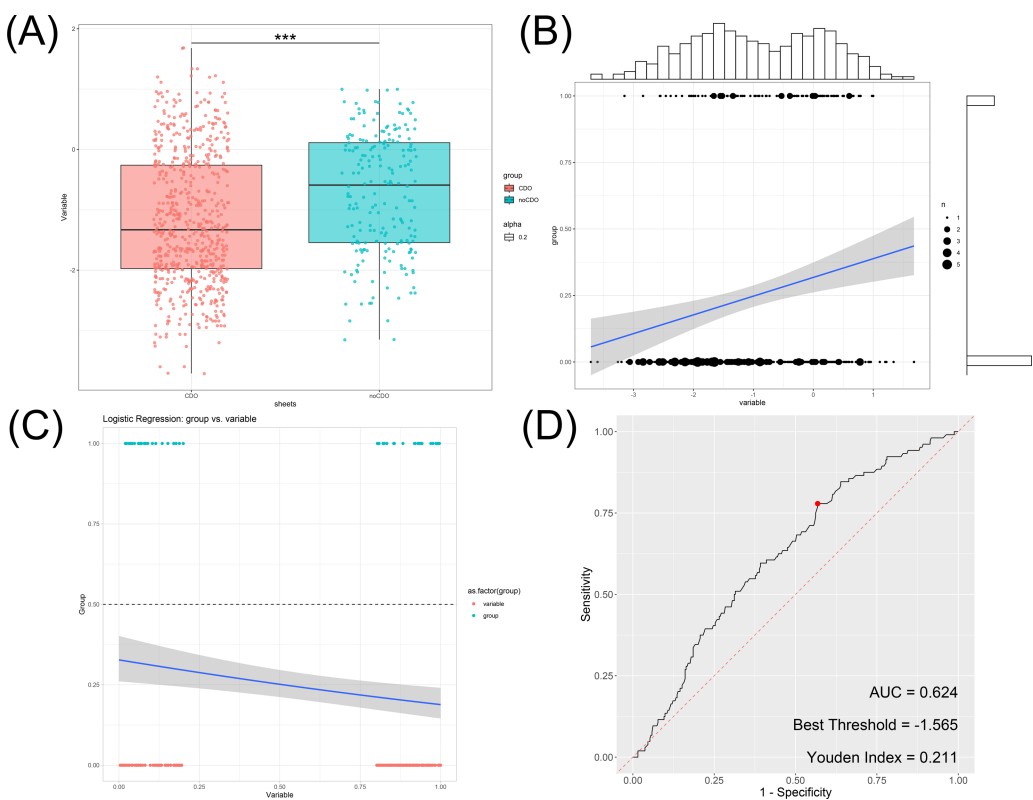

**Figure 1** **Association between bone mineral density and incidence of cardiovascular and cerebrovascular diseases.** (A) Box plot of bone mineral density in cardiovascular disease group and non-cardiovascular disease. (B) The correlation analysis of bone mineral density and cardiovascular disease. (C) The logistic regression analysis of bone mineral density and cardiovascular disease. (D) ROC analysis of bone mineral density and cardiovascular disease.

certain connection between osteoporosis and cardiovascular and cerebrovascular diseases, with certain similarities in pathophysiological mechanism and risk factors including heredity, senescence, unhealthy diet and living habits, endocrine disorder, smoking, lack of exercise, alcohol abuse, *etc.* (*Aggarwal et al., 2023*; *Manubolu et al., 2023*). Studies at home and abroad have shown that bone mineral density as a risk factor of the occurrence and death of cardio-cerebrovascular events can better predict the occurrence and progression of diseases than traditional risk factors such as hyperlipidemia and smoking (*Asadi et al., 2022*; *Akin & Altun, 2022*). Therefore, more and more scholars are concerned and studying the relationship between osteoporosis and cardiovascular and cerebrovascular diseases, which is of great significance for prevention and treatment of diseases.

*Chen et al. (2023)* have shown that the greatly increased risk of hip fracture in hypertension patients exists in both short-term and long-term courses of hypertension. The results of this study displayed that there were statistical differences in blood pressure between the osteoporosis group and the non-osteoporosis group ($P < 0.05$), and there was a correlation between hypertension and bone mineral density T-score. Logistic regression analysis showed that hypertension was a risk factor of osteoporosis. The reason may be that

the fluctuation of blood pressure will increase the risk of falls, accelerate urinary calcium excretion, and reduce the supply of bone blood vessels in hypertension patients, which greatly increases the incidence of osteoporosis, suggesting that it is necessary to prevent the occurrence of osteoporosis in the clinical treatment of hypertension. The findings from this study underscore the importance of preventing osteoporosis during the clinical treatment of hypertension. Additionally, it is worth noting that certain hypertension drugs, such as thiazide diuretics, have shown efficacy in treating osteoporosis. Therefore, when managing hypertension in patients with concurrent osteoporosis, careful consideration should be given to the thoroughly evaluate how the antihypertensive drugs prescribed may affect the bone health of these patients (*Fuggle et al., 2020*).

Our findings revealed significant associations between osteoporosis and various biochemical indexes. In the logistic regression analysis, elevated levels of triglyceride, total cholesterol, and low-density lipoprotein (LDL) were positively associated with the presence of osteoporosis. Conversely, higher levels of high-density lipoprotein (HDL) and homocysteine were negatively associated with osteoporosis. These results suggest that lipid profiles and homocysteine levels may influence bone health and density.

Correlation analysis further supported these findings by demonstrating significant associations between bone mineral density (BMD) and biochemical indexes. Triglyceride levels showed a negative correlation with BMD, while total cholesterol and HDL levels exhibited positive correlations. LDL levels had a weak negative correlation with BMD, and homocysteine levels exhibited a strong negative correlation. These observations indicate that lipid profiles and homocysteine play a potential role in influencing bone health.

Furthermore, we observed a significant difference in BMD and the incidence of cardiovascular disease between the osteoporosis and non-osteoporosis groups. The osteoporosis group had significantly lower BMD and a higher incidence of cardiovascular disease compared to the non-osteoporosis group. This finding suggests a potential correlation between osteoporosis and an increased risk of cardiovascular disease.

A correlation and regression analysis between BMD and cardiovascular disease incidence revealed a significant positive relationship. Individuals with lower BMD, particularly those in the osteoporosis group, had a higher risk of developing cardiovascular disease. Logistic regression analysis confirmed this correlation, providing an odds ratio of 0.477 (95% CI [0.304–0.746]). To assess the utility of BMD in identifying cardiovascular disease, we conducted receiver operating characteristic (ROC) analysis. The area under the curve (AUC) was found to be 0.624, indicating that BMD has moderate discriminatory power in distinguishing the occurrence of cardiovascular disease. The Youden index was calculated to be 0.211, and the optimal threshold value for BMD was determined to be −1.565.

These findings highlight the potential value of BMD as a diagnostic tool in identifying individuals at risk of cardiovascular disease. While the exact mechanisms underlying the observed associations between osteoporosis, biochemical indexes, BMD, and cardiovascular disease require further investigation, our study provides important preliminary evidence on the interconnections between these aspects of health.

The mechanism may be that blood lipid as an important fatty composition in atherosclerotic plaques can competitively inhibit the combination of plasminogen and

platelets, prevent fibrinolytic process and thrombolysis, promote thrombosis formation, and cause cardiovascular and cerebrovascular diseases (*Hu et al., 2019*). Lipid oxides are also associated with the formation of atherosclerosis, inhibiting the differentiation of osteoblasts. The elderly usually have a high pulse pressure and more distinct atherosclerosis. Intravascular nitric oxide plays an important role in osteoblasts and bone transformation. Vascular endothelium is damaged and the release of nitric oxide is reduced during atherosclerosis (*Zhu et al., 2019*). Hcy, widely used in clinic, is a sulphur amino acid, an important intermediate product in the metabolic process of methionine and cysteine, and a risk factor of cardiovascular and cerebrovascular diseases. Abnormal Hcy level leads to arteriosclerosis, and after the combination with hypertension, it is more likely to cause cerebral arteriosclerosis with transient cerebral ischemia, cerebral infarction and cerebral hemorrhage. *Martí-Carvajal et al. (2009)* has shown that elevated Hcy level degrades the extracellular matrix induced by metalloproteinases, reduces blood flow within the bone, and causes the development of bone diseases through oxidative stress-mediated mechanisms. The results of this study revealed that the osteoporosis group had significantly higher Hcy level than the non-osteoporosis group ($P < 0.05$). Bone mineral density T-score was correlated with Hcy level, which was the influencing factor of osteoporosis in patients with cardiovascular and cerebrovascular diseases, suggesting that Hcy level can be used as an important testing index of cardiovascular and cerebrovascular diseases combined with osteoporosis.

Osteoporosis, as a common degenerative disease in the elderly, is difficult to restore normal bone structure once the loss of bone substance occurs (*Syu et al., 2022*). In everyday life, it is crucial for individuals, particularly those with hypertension, coronary heart disease, and cerebral infarction, to proactively manage their health. This includes early detection and intervention of potential health issues, timely medical treatment when symptoms arise, and active prevention of osteoporosis. Controlling blood pressure, managing blood lipids, and monitoring other relevant biochemical markers are beneficial not only for cardiovascular and cerebrovascular health but also for preventing the reduction of bone mass (*Gilbert et al., 2022*). For hypertension patients, thiazide diuretics have been shown to reduce calcium excretion and maintain bone density (*Martí-Carvajal et al., 2009*). Optimizing vitamin D status may also provide benefits for bone and cardiovascular health (*Park et al., 2022b*; *Cipriani et al., 2022*). Osteoporosis screening should be considered in hypertensive patients to enable early detection and treatment. At the same time, medical staff should pay extra attention to the screening of osteoporosis, carry out knowledge education, and tell about harms and prevention measures to patients with cardiovascular and cerebrovascular diseases, which is of great significance for early prevention and control of osteoporosis (*Tang et al., 2022*).

It is worth noting that the present findings are based on a cross-sectional study design, limiting our ability to establish causality or determine long-term prognostic implications. Future longitudinal studies are warranted to validate our results and explore the temporal associations between osteoporosis, biochemical profiles, BMD, and cardiovascular outcomes. Additionally, studies investigating the underlying mechanisms linking these parameters would enhance our understanding of the pathophysiology and potential

therapeutic strategies for managing both osteoporosis and cardiovascular disease. Besides, confounding variables including age, sex, smoking status, and medication use should be accounted for in future longitudinal analyses to accurately delineate the relationships between osteoporosis, biomarkers, and cardiovascular disease. Future studies could focus on elucidating the underlying mechanisms that link osteoporosis, biochemical indices, BMD, and cardiovascular disease. It is important to conduct longitudinal studies to establish causality, determine the temporal associations and time-dependent patterns between these factors, and assess the impact of confounding variables. Additionally, research should aim to investigate the potential use of BMD as a diagnostic tool for identifying individuals at higher risk for cardiovascular disease, explore the mechanisms of specific biomarkers, and address specific therapeutic targets. By addressing these research questions, future studies can deepen our understanding of the complex interplay between osteoporosis, biochemical indices, BMD, and cardiovascular disease, and contribute to the development of effective prevention and treatment strategies.

## CONCLUSIONS

This study provides compelling evidence supporting the associations between osteoporosis, biochemical indices, BMD, and cardiovascular disease. Our results suggest that abnormalities in lipid profiles and homocysteine levels may contribute to the development of osteoporosis. Furthermore, individuals with lower BMD, especially those with osteoporosis, seem to have an increased risk of cardiovascular disease. Notably, BMD demonstrates potential as a diagnostic tool to identify individuals at risk for cardiovascular disease. Additional research is warranted to elucidate the underlying mechanisms and determine the clinical implications of these relationships.

### Funding
The authors received no funding for this work.

### Competing Interests
The authors declare there are no competing interests.

### Author Contributions
- Xiaoying Hu conceived and designed the experiments, performed the experiments, analyzed the data, prepared figures and/or tables, authored or reviewed drafts of the article, and approved the final draft.
- Shucan Ma conceived and designed the experiments, authored or reviewed drafts of the article, and approved the final draft.
- Liman Chen performed the experiments, analyzed the data, authored or reviewed drafts of the article, and approved the final draft.
- Chunhui Tian performed the experiments, prepared figures and/or tables, and approved the final draft.

- Weiwei Wang conceived and designed the experiments, analyzed the data, prepared figures and/or tables, and approved the final draft.

## Human Ethics

The following information was supplied relating to ethical approvals (i.e., approving body and any reference numbers):

All samples obtained in this study were approved by the ethics committee of the Harrison International Peace Hospital and abided by the ethical guidelines of the Declaration of Helsinki.

## Data Availability

The raw data is available in the Supplemental File.

## Supplemental Information

Supplemental information for this article can be found online at http://dx.doi.org/10.7717/peerj.16546#supplemental-information.

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
