# Peer review of "Association between osteoporosis and cardiovascular disease in elderly people: evidence from a retrospective study"

_PeerJ, doi:10.7717/peerj.16546_

## Round 0.1 · original submission · Major Revisions

Major issues:

1. This study retrospectively analyzed the clinical data of patients with coronary heart disease, hypertension, and cerebral infarction. It would be beneficial to provide more information about how these specific conditions were diagnosed in the patients.

2. The study used the T-score as the judgment standard of bone mass, but it does not provide information on how the T-score was calculated or the specific equipment used to measure.

3. The authors mention that the patients had "certain cognitive functions" as part of the inclusion criteria, but it does not specify what these cognitive functions are or how they were assessed.

4. Figure 1 is not clearly labeled in the text. Authors need to clearly state which results correspond to A, B, C and D of Figure 1.

5. The resolution of all parts of Figure 1 is relatively poor, and the author needs to improve the quality of Figure 1, e.g., by increasing the size of the fonts.

Minor issues:

1. [The incidence of osteoporosis are] should be [The incidence of osteoporosis is].

2. [more scholars concern and study] should be [more scholars are concerned and studying].

3. The article, a or the, is regularly missed.

4. [have showed] should be [have shown].

5. [greatly increase] should be [greatly increases].

6. [reduces bone blood flow, and cause] should be [reduces bone blood flow, and causes].

7. Language editing is needed.

**Language Note:** The Academic Editor has identified that the English language must be improved. PeerJ can provide language editing services - please contact us at [email protected] for pricing (be sure to provide your manuscript number and title). Alternatively, you should make your own arrangements to improve the language quality and provide details in your response letter. – PeerJ Staff

Reviewer 1 ·

Basic reporting

The manuscript provides a compelling analytical contribution in the interrelated fields of osteoporosis and cardiovascular disease, suggesting valuable correlations between biochemical profiles, Bone Mineral Density (BMD), and cardiovascular disease. Employing technically sound and comprehensive English, the authors present a logically structured examination of a vital intersection in medical research. The introduction and background are adequately described, shedding light on how this study is relevant to the broader field. However, the gap in knowledge being filled by this study is implicitly stated and could be better highlighted to underline the importance of their work. On the technical front, the research uses sound statistical measures, but a more detailed description of why these specific techniques were chosen for data interpretation would strengthen the manuscript. Although inferred, the manuscript could lodge details on data availability, robustness, statistical soundness, and control measures, to sharply enhance its transparency and replicability. Overall, despite a few gaps, the research provides a thought-provoking context on the relationships between osteoporosis and cardiovascular diseases and impresses on the pressing need for further studies to explicate these relations fully.

Experimental design

1. More details about what Bone Mineral Density (BMD) is would be useful for readers unfamiliar with the concept.
2. Explain why ROC analysis was chosen and how the optimal threshold value for BMD was determined.

Validity of the findings

1. A detailed and proper explanation of the physiological mechanisms behind the correlation between hypertension, osteoporosis, and cardiovascular diseases needs to be added to enhance understanding.
2. Clarify the biochemical indexes used in the study to give the reader better understanding.
3. When discussing the correlations between various biochemical indicators and osteoporosis, take time to interpret what these correlations might mean in a biological context.
4. Provide specific strategies related to interventions and treatment to prevent osteoporosis for hypertension patients, and how to control related biochemical indicators.
5. Discuss the most effective methods for screening osteoporosis and how these methods can be integrated into clinical practice.
6. While you mention that causality cannot be established due to the study design, please provide a detailed explanation of other limitations that may impact the results of your study.

Additional comments

1. The introduction should include more detailed information about the current understandings and gap in the literature to justify the study. Mention key literature reviews related to the topic.
2. You list several risk factors, such as heredity, senescence, etc. Some further explanation of how these contribute to osteoporosis and cardiovascular disease could be beneficial.

Reviewer 2 ·

Basic reporting

This manuscript endeavors to elucidate the intricate connection between osteoporosis and cardiovascular diseases by exploring the impact of biochemical indices and Bone Mineral Density (BMD). Regrettably, it currently lacks critical details, impeding its ability to contribute effectively to the field. The methodologies employed in the study are insufficiently detailed, rendering replication of the research a challenge. To rectify this, a comprehensive, step-by-step description of the methods used is essential to facilitate validation through independent testing. Moreover, the manuscript exhibits a noticeable deficiency of information concerning the accessibility, stability, and statistical integrity of the base data. An unambiguous statement addressing these factors and outlining the control measures implemented to verify data authenticity should be incorporated. Nevertheless, this study deftly interlaces multiple facets of the entangled osteoporosis and heart disease narrative, thereby creating a solid foundation for prospective research.

Experimental design

I. The proposed mechanism needs to be presented in a clear and concise way.
II. Please include a thorough description of your methods, including selection of participants, data collection, and analysis.

Validity of the findings

I. More evidence to support the connection between osteoporosis and cardiovascular diseases would strengthen the argument.
II. The significance of elevated homocysteine (Hcy) levels in relation to bone diseases needs further explanation.
III. The manuscript would benefit from additional discussion on how the results can be applied clinically. What do these findings mean for patients with cardiovascular diseases or osteoporosis and their healthcare providers?
IV. Make sure your conclusions are concise, clear, and directly supported by the results of your study.

Additional comments

Maintain consistency in your use of terminology to prevent confusion. For example, keep the use of 'cardio-cerebrovascular diseases' or 'cardiovascular and cerebrovascular diseases' consistent.

Reviewer 3 ·

Basic reporting

This manuscript presents important findings contributing to the understanding of the interplay between osteoporosis, biochemical indexes, BMD, and cardiovascular disease. However, improvements in providing comprehensive methodological details, demonstrating data availability and robustness, explicating statistical soundness, and clarifying control measures would strengthen the overall quality and impact of the study.

Experimental design

1) Specify the reason for measuring bone mineral density at specific locations and provide additional information about the measurement technique. Explain why the lumbar vertebrae, femur neck, Ward’s triangle, and femoral trochanteric region were chosen for measurement.
2) Specify the significance level used for statistical tests. State the alpha level (e.g., p < 0.05) used to determine statistical significance.
3) The paragraph mentions "certain connection" between osteoporosis and cardiovascular and cerebrovascular diseases but does not provide any specific details. Expand on this point by explaining the shared risk factors or mechanisms that link these conditions.

Validity of the findings

4) When reporting the significant associations between osteoporosis and the biochemical indexes, provide the effect sizes or measures of association (e.g., odds ratios, correlation coefficients) to quantify the strength of these associations.
5) Explain how the logistic regression analysis was conducted, including the independent variables included in the model and any potential confounding factors that were accounted for.

Additional comments

6) Consider Confounding Factors and Control Measures: Explicitly identify the confounding variables that were controlled for during the analysis and discuss any limitations or potential biases. This will ensure a more accurate interpretation of the results.

---

## Round 0.2 · Minor Revisions

Other issues that need to be further addressed:
1. The authors use the term "Hcy" without defining it first.
2. "once uncomfortable symptoms appear" is somewhat vague. It would be more precise if the authors could specify what these symptoms might be.
3. The sentence "Some hypertension drugs have certain effects on the treatment of osteoporosis, such as thiazide diuretics, so it is necessary to carefully consider the mechanism of drugs used in the clinical treatment of hypertension combined with osteoporosis." is a bit confusing.
4. "bone blood flow" is a bit unclear. It would be clearer if the authors could specify whether they are referring to blood flow within the bone or to the bone.
5. "bone diseases through oxidative stress" could be rephrased for clarity.
6. "the related population should achieve early discovery, intervention and treatment" is somewhat vague. It would be clearer if the authors could specify who the "related population" is and how they should be discovering, intervening, and treating early. [In daily life, - - - -, and bone mass reduction] should be re-written.

Reviewer 1 ·

Basic reporting

The author has made good revisions to this article and carefully responded to my comments. I have no further comments.

Experimental design

The author has made good revisions to this article and carefully responded to my comments. I have no further comments.

Validity of the findings

The author has made good revisions to this article and carefully responded to my comments. I have no further comments.

Additional comments

Your revised manuscript is acceptable for publication.

Reviewer 2 ·

Basic reporting

Good.

Experimental design

Expand on the rationale for the inclusion criteria, especially the criteria related to cognitive function and high degrees of cooperation. Explain why these criteria are important for the study and how they ensure reliable data collection.

Validity of the findings

Good.

Additional comments

1.You've mentioned the need for future studies. Can you provide more detail on what specific questions future research should seek to answer?
2.The manuscript should be proof-read to ensure there are no typographical errors and that the language used is formal, scientific and in third person perspective.

Reviewer 3 ·

Basic reporting

After the author's modifications, the structure of the article became clear and clear, using professional English throughout.
References provide sufficient background. The overall modification is good.

Experimental design

After modification, the definition of the research question has become clear, relevant, and meaningful. It explains how research can fill identified knowledge gaps. The described method has sufficient details and information for replication.

Validity of the findings

The data provided by the author is robust, statistically reliable, and controllable.
The conclusion section is fully stated and well revised.

Additional comments

After careful and rigorous editing by the author, I believe that this article meets the publication standards of the magazine.

---

## Round 0.3 · accepted · Accept

My concerns were adequately addressed, and I think this revised version could be considered for publication in this journal.